# CD163-Mediated Small-Vessel Injury in Alzheimer’s Disease: An Exploration from Neuroimaging to Transcriptomics

**DOI:** 10.3390/ijms25042293

**Published:** 2024-02-14

**Authors:** Yuewei Chen, Peiwen Lu, Shengju Wu, Jie Yang, Wanwan Liu, Zhijun Zhang, Qun Xu

**Affiliations:** 1Health Management Center, Renji Hospital of Medical School, Shanghai Jiao Tong University, Shanghai 200127, China; chenyuewei_js@163.com (Y.C.); lupeiwen0803@163.com (P.L.); liuwanwan@renji.com (W.L.); 2Department of Neurology, Renji Hospital of Medical School, Shanghai Jiao Tong University, Shanghai 200127, China; 3Renji-UNSW CHeBA (Centre for Healthy Brain Ageing of University of New South Wales) Neurocognitive Center, Renji Hospital of Medical School, Shanghai Jiao Tong University, Shanghai 200127, China; 4School of Biomedical Engineering, Shanghai Jiao Tong University, Shanghai 200030, China

**Keywords:** CD163, Alzheimer’s disease, white-matter hyperintensities, mild cognitive impairment, transcriptome

## Abstract

Patients with Alzheimer’s disease (AD) often present with imaging features indicative of small-vessel injury, among which, white-matter hyperintensities (WMHs) are the most prevalent. However, the underlying mechanism of the association between AD and small-vessel injury is still obscure. The aim of this study is to investigate the mechanism of small-vessel injury in AD. Differential gene expression analyses were conducted to identify the genes related to WMHs separately in mild cognitive impairment (MCI) and cognitively normal (CN) subjects from the ADNI database. The WMH-related genes identified in patients with MCI were considered to be associated with small-vessel injury in early AD. Functional enrichment analyses and a protein–protein interaction (PPI) network were performed to explore the pathway and hub genes related to the mechanism of small-vessel injury in MCI. Subsequently, the Boruta algorithm and support vector machine recursive feature elimination (SVM-RFE) algorithm were performed to identify feature-selection genes. Finally, the mechanism of small-vessel injury was analyzed in MCI from the immunological perspectives; the relationship of feature-selection genes with various immune cells and neuroimaging indices were also explored. Furthermore, 5×FAD mice were used to demonstrate the genes related to small-vessel injury. The results of the logistic regression analyses suggested that WMHs significantly contributed to MCI, the early stage of AD. A total of 276 genes were determined as WMH-related genes in patients with MCI, while 203 WMH-related genes were obtained in CN patients. Among them, only 15 genes overlapped and were thus identified as the crosstalk genes. By employing the Boruta and SVM-RFE algorithms, CD163, ALDH3B1, MIR22HG, DTX2, FOLR2, ALDH2, and ZNF23 were recognized as the feature-selection genes linked to small-vessel injury in MCI. After considering the results from the PPI network, CD163 was finally determined as the critical WMH-related gene in MCI. The expression of CD163 was correlated with fractional anisotropy (FA) values in regions that are vulnerable to small-vessel injury in AD. The immunostaining and RT-qPCR results from the verifying experiments demonstrated that the indicators of small-vessel injury presented in the cortical tissue of 5×FAD mice and related to the upregulation of CD163 expression. CD163 may be the most pivotal candidates related to small-vessel injury in early AD.

## 1. Introduction

Alzheimer’s disease (AD) and cerebral small-vessel disease (CSVD) rank as the predominant causes of dementia [1]. Recent data prognosticates that, by 2050, the prevalence of dementia will triple worldwide. Notably, this estimate increases threefold when considering a biological definition of AD, rather than relying on clinical criteria [2]. AD, a neurodegenerative disorder, manifests through the abnormal accumulation of amyloid beta (Aβ) plaques and neurofibrillary tangles (NFTs), along with progressive neuronal loss, culminating in brain atrophy, cognitive decline, and behavioral disturbances [3]. The continuum of AD, stretching over a period of 15–25 years, can be present without any overt symptoms via a stage of mild cognitive impairment (MCI) leading up to dementia [2]. The stage of MCI is always considered as early AD, which is a critical period for disease modification. In addition, CSVD, encompassing white-matter lesions (WML), an enlarged perivascular space (EPVS), cortical superficial siderosis, lacunes, and cerebral microbleeds [4], plays a pivotal role not only in the direct etiology of vascular dementia but is also prominently encountered within AD brains [5,6]. The two-hit vascular hypothesis for AD proposes that primary damage to the brain microcirculation (hit one) initiates a non-amyloidogenic pathway leading to Aβ accumulation (hit two). These peptides, in turn, exert vasculotoxic and neurotoxic effects [7].

White-matter hyperintensities (WMH) have traditionally been viewed as one of the most prevalent neuroimaging features of small-vessel injury, especially for CSVD. Many neuroimaging studies have shown a relationship between AD pathologies and WMH [8,9]. For instance, a recent study emphasized the clinical relevance of WMHs in AD, especially posterior WMHs, and most notably, splenium of the corpus callosum (S-CC) WMH [10]. Another study pointed that posterior WMHs might be related to degenerative mechanisms secondary to AD pathology, while anterior WMH could be associated with both CSVD and degenerative mechanisms [11], indicating that the pathological mechanism of small-vessel injury in AD may be different from that in sporadic CSVD. Compared with traditional structural MRI sequences, indices extracted from diffusion tensor imaging (DTI) more sensitively detected white-matter microstructure damages due to small-vessel injury [12,13,14].

Recently, bioinformatics approaches have been developed to explore the underlying mechanisms of diseases. Feature-selection algorithms, including the Boruta algorithm and support vector machine recursive feature elimination (SVM-RFE) algorithm, have already been applied in transcriptomic, proteomic, and metabolomic analyses for targeted substance screening [15,16,17,18]. However, there have been few studies on the mechanisms of small-vessel injury in AD based on these technologies. Therefore, the primary objective of this study is to investigate the distinct mechanism of small-vessel injury in AD, by utilizing neuroimaging with transcriptome analysis, immunological explorations, and further experimental verification.

AD datasets sourced from the ADNI database were employed for comprehensive systemic analysis. First, logistic regression models were constructed to evaluate the contributions of WMH and cerebral atrophy in distinct stages of AD. Second, transcript-level differential analysis was conducted to obtain DEGs. Third, hub genes were identified using maximal clique centrality (MCC) and the molecular complex detection (MCODE). Fourth, the Boruta algorithm and SVM-RFE algorithm were used to obtain feature-selection genes. Machine Learning models were then employed to verify the robustness of these feature-selection genes. Fifth, the correlation analyses between feature-selection genes, the abundance of infiltrating immune cells, and neuroimaging features were subsequently conducted to illustrate the small-vessel injury during early AD from immunological perspectives and establish connections in multidimensional data. Finally, the results were further validated through cellular and animal experiments. The detailed analytic workflow chart is shown in Figure 1.

## 2. Results

### 2.1. WMH and Cerebral Atrophy Both Contributed to Early Stage of Alzheimer’s Disease

Demographic characteristics, neuropsychological tests, and neuroimaging findings of the different cognitive groups are shown in Table 1. Standardized total cerebrum brain volume to total cerebrum cranial volume (TCB_TCV) and standardized segmented total hippocampi volume to TCV (T_hippo_TCV) significantly decreased sequentially in cognitively normal (CN) patients, MCI, AD (all *p* < 0.05). Meanwhile, indicators of WMH did not exhibit significant differences among the groups (all *p* > 0.05). Differences in DTI indices, including fractional anisotropy (FA), and mean, axial, and radial diffusivity (MD, AxD, RD) values, were observed among the three groups, especially for the following fiber bundles: tapetum, bilateral splenium of the corpus callosum, bilateral fornix, posterior thalamic radiation_left, fornix (cres)/Stria terminalis_left (FDR-adjusted *p* < 0.05). These were consistent with the vulnerable regions reported in previous studies [14] and may be associated with AD pathology (Figure 2A,B, Appendix A).

Multivariate logistic regression models revealed that TCB_TCV was associated with both MCI and dementia (all *p* < 0.05), while the natural logarithm of standardized WMH to TCV (ln_WMH_TCV) was only related to MCI (*p* < 0.05). Joint diagnostic effectiveness for TCB_TCV and ln_WMH_TCV in MCI yielded an area under the curve (AUC) value of 0.682 (TCB_TCV: OR = 0.882 (0.804~0.966) *p* < 0.01; ln_WMH_TCV, OR = 1.278 (1.053~1.559) *p* < 0.05) (Table 2, Figure 2C).

### 2.2. Differentially Expressed Genes (DEGs) and Functional Enrichment Analysis of WMH-Related Genes in MCI

Demographic characteristics, neuropsychological tests, and neuroimaging indices for either CN patients or those with MCI stratified by the median of ln_WMH_TCV are shown in Appendix A. WMH+ indicates where the values of ‘ ln_WMH_TCV’ were above the median, while WMH– indicates values below the median. A total of 109 upregulated DEGs and 94 downregulated DEGs were identified in the CN_WMH+ group, while 112 upregulated DEGs and 164 downregulated DEGs were identified in the MCI_WMH+ group (Figure 3A,B). Only 15 genes overlapped between the CN_WMH+ and MCI_WMH+ groups (Figure 3C). These results suggested that the mechanism of small-vessel injury in patients with MCI (due to AD pathology) differed from that in CN patients.

The results of the gene set enrichment analysis (GSEA) of the Kyoto Encyclopedia of Genes and Genomes (KEGG) pathways further revealed that the WMH-related genes in MCI were mainly enriched in pathways related to infection, immunity, and metabolism, including “lysosomes”, “phagosomes”, “neutrophil extracellular trap formation”, “Influenza A” and “retinol metabolism” (Figure 3D,E). GSEA of gene ontology (GO) yielded similar outcomes as well (Appendix A).

### 2.3. PPI Network Construction of WMH-Related Genes in MCI

Based on the STRING results, a PPI network of WMH-related genes was constructed using 269 nodes and 285 edges in MCI (Figure 4A). A total of 19 genes and 30 genes were identified as being closely related genes using the MCODE (Figure 4B–D) and cytoHubba plugins (MCC) (Figure 4E), respectively. Interestingly, there were 15 hub genes that overlapped in both hub genes selection processes (Figure 4F).

### 2.4. Feature-Selection Genes from WMH-Related Genes in MCI

Based on the MCI_WMH-related gene set, 15 feature-selection genes were selected using the Boruta algorithm (ALDH2, MAST2, TCN2, CD163, ALDH3B1, CCDC170, ZNF23, MIR22HG, CCR6, NLRP3, PNPLA8, FOLR2, DTX2, EPHX1, PLEK) and 22 feature-selection genes were selected using the SVM-RFE algorithm. The results of feature-selection genes based on the Boruta and SVM-RFE algorithms are shown in Figure 5A,B. A total of seven variables were selected by both algorithms (Figure 5C), namely CD163, ALDH3B1, MIR22HG, DTX2, FOLR2, and ALDH2, ZNF23. Using the above feature-selection genes, seven machine learning models were established to predict the degree of WMH in the MCI group. The receiver operating characteristic (ROC) curve of each model showed predictive effectiveness, represented by the AUC value (Figure 5D). The gradient boosting machine (GBM) (AUC = 0.741) was the best model for predicting the degree of WMH in MCI groups, followed by random forest (RF) (AUC = 0.727) and k-nearest neighbors (KNN) (AUC = 0.705). The predictive effectiveness of seven machine learning methods in the training set is illustrated in Appendix A. In summary, these seven genes were identified as the feature-selection genes and demonstrated their potential as biomarkers for predicting the degree of small-vessel injury in patients with MCI. From the expression values of boxplots, CD163, FOLR2, ALDH3B1, DTX2, and ALDH2 were highly expressed in the MCI group with severe small-vessel injury (Figure 5E).

### 2.5. Immune Landscape and Feature-Selection-Genes Correlation Analysis

The boxplot of infiltrating immune cell types revealed that the abundances of regulatory T cells, macrophages, monocytes, natural killer T cells, myeloid-derived suppressor cells, central memory CD8 T cells, and immature dendritic cells were much more increased in the MCI_WMH+ group than in the MCI_WMH– group, whereas the activated B cells were significantly reduced (Figure 6A). The correlation between feature-selection gene expression and the abundance of infiltrating immune cell types showed that most immunocytes positively connected with the seven feature-selection genes (Figure 6B). These results implied that the inflammatory components play an essential role in small-vessel injury in MCI, and most feature-selection genes are primarily associated with the positive regulation of immune function.

The feature-selection genes, especially CD163 and FOLR2, exhibited significant correlations with FA values. This correlation was related to the fragility of small vessels in specific brain regions, with a tendency towards the left side, including left posterior thalamic radiation and the left fornix, considered as posterior WMHs (Figure 6C, Appendix A).

Based on the results of PPI and machine learning, CD163 and FOLR2 were determined as the key WMH-related genes of MCI (Figure 6D). The significant correlations of either CD163 or FOLR2 with ln_WMH_TCV are shown in Figure 6E.

### 2.6. AD Pathology Resulted in Small-Vessel Injury and Elevated CD163 Expression

To further validate the alterations in the expression of CD163 following small-vessel injury in the context of AD pathology, we performed experiments using 6-month-old 5×FAD mice. Immunofluorescence results revealed the existence of cortical EPVS, which is an important indicator of small-vessel injury (Figure 7A–E). Real-time quantitative polymerase chain reaction (RT-qPCR) results from cortical tissues exhibited the upregulation of CD163 expression (Figure 7F). RT-qPCR analysis of rat primary microglia showed that CD163 expression increased after Aβ treatment (Figure 7G,H). This suggested that AD pathologies, especially Aβ plaques, may contribute to small-vessel injury and upregulate the expression of CD163.

## 3. Discussion

There has been limited research centered around transcriptomic strategies on the mechanisms of small-vessel injury in AD. Through comprehensive analyses of clinical data, reviews of the relevant literature, and the verification experiments, our study suggested a distinct mechanism of small-vessel injury in early-stage AD.

From both clinical and neuroimaging perspectives, although WMHs did not show significant differences among the three groups, sensitive DTI indices, such as FA values, identified vulnerable regions associated with small-vessel injury in the AD continuum. We also identified that WMHs, an indicator of small-vessel injury associated with the diagnosis of early-stage AD, and TCB, which represents cerebral atrophy, can associated with the entire course of AD. Small-vessel injury played a potential role in the initial phases of the disease, whereas degenerative factors predominated in all stages of the disease [19,20]. These results also provide evidence to support the “two-hit vascular hypothesis” of AD etiology [21].

After bioinformatics analyses, we found that the gene expression of small-vessel injury in patients with MCI were quite different from that in CN patients. Increasing evidence shows that WMHs may predict the probability for diagnosing MCI [19,22], and the period of MCI is a crucial period for disease intervention within the AD continuum. This intrigues us and we subsequently wish to focus on small-vessel injury in the context of MCI. Seven feature-selection genes (CD163, FOLR2, ALDH3B1, MIR22HG, DTX2, ALDH2, and ZNF23), out of 276 WMH-related genes in MCI, were selected by the Boruta algorithm and the SVM-RFE algorithm and further validated as the predictable biomarkers of WMH through seven machine learning risk prediction models. Notably, they were highly correlated with the immune cells, as well as the immune-related pathway, indicating a potential role through immune-related biological pathways in the development of small-vessel injury in early AD. Finally, CD163 and FOLR2 were identified as the pivotal genes.

CD163, a glycoprotein within class B of the scavenger receptor cysteine-rich superfamily, primarily participates in iron metabolism, inflammation, and immune responses. Traditionally, its expression was thought to be restricted to perivascular and meningeal macrophages (Appendix A) [23,24]. However, CD163 is also expressed in the microglia in various nervous system diseases, such as AD, Parkinson’s disease (PD), HIV-encephalitis, multiple sclerosis, and head injury tissue [24,25,26,27]. Notably, CD163 exhibits heightened expression in AD, particularly in the frontal and occipital cortices, with co-localization with Aβ; in addition, it was found at a higher density around compromised blood vessels [24,28]. Our experimental results also reveal the presence of small-vessel injury in the cortex of 5×FAD mice, accompanied by an elevated expression of CD163. Additionally, exposure to Aβ induces the increase in CD163 expression in primary microglial cells. Research on human coronary artery plaques has demonstrated that the presence of CD163^+^ macrophages were associated with increased expression of endothelial vascular cell adhesion molecule (VCAM), angiogenesis, inflammatory cell recruitment, and high microvascular permeability through the CD163/HIF1α/VEGF-A pathway [29].

Increased CD163 expression was found in the MCI_WMH+ group by our analysis. Therefore, it is reasonable to hypothesize that early hypoperfusion may hinder Aβ clearance, thereby promoting Aβ deposition and upregulating CD163 expression. This upregulation of CD163 expression, driven by the CD163/HIF1α/VEGF pathway, initiates non-functional angiogenesis, chronic inflammatory cell recruitment, and heightened vascular permeability, and exacerbates Aβ deposition. These processes create a detrimental feedback loop, ultimately resulting in development of the MCI stage. Certainly, these hypotheses need further validation by modulating the expression of CD163 in microglial cells to investigate its impact on the blood-brain barrier (BBB).

The GO annotations for FOLR2, which encode a member of the folate receptor (FOLR) family, include folic acid binding and folic acid transmembrane transporter activity. Folate deficiency is associated with cerebrovascular diseases, neurological diseases, and mood disorders [30]. Given the reported protection exerted by folic acid against oxidative stress, resulting from exposure to amyloid beta [31], the observed upregulation of folate receptors and folate binding could represent a response to the increased intracellular vitamin need. Therefore, lower circulating serum folate may be attributable to increased folate binding in peripheral tissues such as via fibroblasts [32]. In addition, the upregulation of FOLR2 may contribute to an unfavorable vascular phenotypic switch induced by obesity [33]. However, limited evidence exists on the relationship between this gene and cerebral blood vessels, especially small vessels.

It is noteworthy that CD163 and FOLR2 serve as frequently observed biomarkers of M2 macrophages [34,35,36,37], which are well-known for their anti-inflammatory properties [38]. Our investigation also revealed a positive correlation between the expression of CD163 and FOLR2, suggesting an upregulation of anti-inflammatory macrophages in patients with MCI with pronounced cerebral small-vessel injury. An increased abundance of anti-inflammatory macrophages may signify a compensatory response to an elevated immune milieu within the organism. Furthermore, this finding aligns with the heightened macrophage abundance revealed by immune infiltration analysis.

However, ALDH2 and ALDH3B1 are both members of the aldehyde dehydrogenase (ALDH) protein family, critical for detoxifying aldehydes. ALDH2, a nuclear gene, is transported and functions in the mitochondrial matrix. Elevated blood ALDH2 expression may indicate a protective response to toxic aldehydes in mitochondria [39]. ALDH2 mRNA expression was significantly higher in late-onset AD than in controls and increased with age in wildtype mice [40]. ALDH3B1 also protects cells from oxidative stress [41] and may have protective roles in various brain diseases including epilepsy [42].

DTX2 is a member of the DELTEX (DTX) family of E3 ubiquitin ligases (comprising five members: DTX1, DTX2, DTX3, DTX3L, and DTX4) in mammals and it is closely related to cell growth, differentiation, apoptosis, signal transduction, and some diseases, including tumors [43,44]. MIR22HG, a well-studied lncRNA, functions as a master regulator in diverse malignancies, playing a critical role in various aspects of carcinogenesis, including proliferation, apoptosis, invasion, and metastasis [45]. ZNF23 induces apoptosis in human ovarian cancer cells [46]. However, these three genes have been the subject of limited research in regard to the nervous system.

There are several limitations of this study that are worthy of mentioning. Firstly, brain MRI and gene expression data were available only in the subsamples, primarily due to limited access. Notably, there is a lack of databases containing comprehensive neuroimaging and gene expression data for the AD continuum. To compensate for this limitation, we randomly divided the dataset into a training set (20%) and a testing set (80%), employing as many as seven machine learning algorithms to create ROC curves and verify the robustness of these feature-selection genes. The predictive effectiveness of the GBM, RF, and KNN algorithms had accuracies exceeding 0.7, confirming the robustness of our results. In addition, we are in the process of constructing an independent AD-cohort dataset to further evaluate the generalizability of these feature-selection genes.

Secondly, we have only validated the coexistence of small-vessel injury and elevated CD163 expression in the context of AD pathology. More systematic experiments, including the modulation of CD163 expression in microglia, are required to investigate its impact and the underlying mechanisms of small-vessel injury in the future. Despite the insufficient evidence from the literature reviews regarding the association between FOLR2 and small-vessel injury, this represents one of the research directions that requires further experimental validation.

Finally, some cells and molecules from the central nervous system (CNS) can traverse the BBB and enter the peripheral blood, which provide insights for the state of the CNS, especially in disease research and monitoring. It is essential to acknowledge that peripheral changes cannot directly and fully reveal the pathological and physiological changes in the brain, which is an inherent limitation when studying central diseases through peripheral approaches.

## 4. Materials and Methods

### 4.1. Description of ADNI Subjects in the Study, Dataset Acquisition, and Data Preprocessing

Brain imaging and gene expression data were obtained from the ADNI database (http://adni.loni.usc.edu; accessed on 3 July 2023), a large dataset established in 2003 to measure the progression of healthy and cognitively impaired participants with brain scans, biological markers, and neuropsychological assessments [47]. Peripheral blood samples were collected and the Affymetrix Human Genome U219 Array (Affymetrix, Santa Clara, CA, USA) was utilized for expression profiling. Tabulation of gene expression profiles from blood RNA and all quality control and normalizations were conducted by ADNI before inclusion in the dataset.

Diagnosis, age, gender, education, cognitive test scores, and the most recent imaging data, including WMH as well as DTI indices extracted from MRI, were obtained from ADNI. Using the conversion between VISCODE and VISCODE2, the intervals between MRI, diagnosis, and blood sample collection were controlled so as to not exceed three months.

TCB and WMH were adopted to represent the severity of cerebral atrophy and small-vessel injuries [48,49,50]. For analytical purposes, Both TCB and WMH were standardized by TCV and subsequently multiplied by 100. WMH_TCV was a natural logarithm transformed to mitigate the impact of left skewness distributions (Appendix A).

Logistic regression models were utilized to estimate the odds ratios (ORs) and 95% confidence intervals (CIs) for AD stages associated with cerebral atrophy and small-vessel injury. Age, gender, education level, and APOE mutation were adjusted before regression and correlation analysis.

### 4.2. MRI Analysis

DTI indices including FA, MD, AxD, and the RD of white matter, were sourced from the ADNI database under the same screening criteria as previously described. The GRETNA toolbox [51] was used to perform a one-way ANOVA on the DTI data of the three groups (CN, MCI, dementia), controlling for age, gender, and education level, and corrected for false discovery rate (FDR). Referring to the “JHU ICBM-DTI-81 White-Matter Labels” fiber bundles label, differential fiber bundles were extracted by employing FMRIB’s Software Library (FSL) [52,53] and subsequently visualized using the MRIcron toolbox.

### 4.3. Differential Gene Expression Analysis

To identify WMH-related genes both in the CN and MCI groups, each group was separately stratified into two subgroups. These subgroups, denoted as the WMH+ and WMH– groups, were distinguished based on the median ln_WMH_TCV. This categorization allowed a comprehensive exploration of WMH-related genetic factors within and between the CN and MCI groups. Differential gene expression analysis between WMH- and WMH+ groups with MCI was undertaken using the *limma* package in R (v4.2) [54]. |log2(fold change) | > 0.12, and *p* < 0.05 were considered as screening thresholds to obtain DEGs. Volcano plots of DEGs were plotted using *ggplot2* in R.

### 4.4. Enrichment Analysis

Functional enrichment analysis was carried out using three domains of gene ontology (GO), including biological process (BP), cellular component (CC), and molecular function (MF). KEGG pathway analysis was adopted to identify the pathways of biological molecular interaction. GSEA, utilizing annotations from both GO and KEGG, was executed with the *clusterProfiler* R package [55]. Visualization of the results was achieved using *gseaplot2*, with significance set at a threshold of *p* < 0.05.

### 4.5. Protein–Protein Interactions (PPIs)

All of the DEGs were imported into the STRING online database (https://cn.string-db.org/; accessed on 12 October 2023), a functional protein association network, assembling all known and predicted proteins. The PPI network interactions file with medium confidence scores ≥ 0.4 was downloaded. The MCC plugin [56] and MCODE plugin in Cytoscape were used for further analysis of the interaction network and hub genes. The criteria of MCODE were set as degree cutoff = 2, node score cutoff = 0.2, k-core = 2, and max depth = 100. And the top subnetworks were shown using the MCODE plugin. Subsequently, MCODE analysis was performed and the top three modules in each DEG’s upregulated and downregulated PPI network were obtained. The hub genes were filtered by the MCC plugin for the top 30 genes.

### 4.6. Identification and Validation of Feature-Selection Genes Using Machine Learning

The dataset’s DEGs were subjected to feature selection using the Boruta algorithm [57] and the SVM-RFE algorithm. Following the execution of these algorithms, we determined the feature-selection genes by identifying the intersection of the selected features, and subsequently integrated them into the machine learning model. Seven machine learning algorithms were used to build models, namely the generalized linear model (GLM), gradient boosting machine (GBM), K-nearest neighbors (KNN), random forest (RF), extreme gradient boosting (XGBoost), support vector machine (SVM), and recursive partition tree (RPART). The data of 304 patients with MCI were randomly divided into the training set (80%) and testing set (20%) according to the ratio of 8:2. In order to assess the robustness of the model, we employed tenfold cross-validation on the training set and repeated it three times. On the training set, seven machine learning algorithms were used to build the models, and the testing set was used to test the effectiveness of the model. The model with the maximum AUC value of the receiver ROC was evaluated as the best model. The R packages used included *Boruta*, *caret*, *e1071*, *pROC*, *XGboost*, and *dplyr.*

### 4.7. Immune Cell Infiltration

The relative abundances of 28 infiltrating immune cells in the ADNI dataset were quantified using the ssGSEA algorithm. Boxplots were drawn to demonstrate the differential abundances of the 28 infiltrating immune cells.

### 4.8. The Correlation Analyses of Feature-Selection Genes

Spearman correlations were calculated for the abundances of 28 infiltrating immune cells and FA values, which represent specific fiber bundles susceptible to microvascular damage in the MCI group, with feature-selection genes, followed by visualization using the *pheatmap* package.

### 4.9. Animal and Cell Experiments

#### 4.9.1. Animal and Cell Model of Alzheimer’s Disease

Animal experiments were approved by the Animal Ethics and Experimentation Committee of Shanghai Jiao Tong University, Shanghai, China, and carried out according to the Guide for the Care and Use of Laboratory Animals: Reporting of In Vivo Experiments (ARRIVE) guidelines [58]. Five-month-old 5×FAD and WT mice were purchased from Aniphe Biolaboratory Inc. (*n* = 5). The mice had free access to food and water ad libitum. Mice were sacrificed one month later (m = 6 month); left brain tissues were obtained for immunofluorescence (IF) staining and right brain tissues for RT-qPCR analysis.

Primary mixed glial cells were prepared from the frontal cortices of grouped male and female post-natal day 2 Sprague Dawley rats from the same litter, purchased from Beijing Vital River Laboratory Animal Technology Co., Ltd. In brief, cerebral cortices were cleaned from all meninges, digested in trypsin, and dissociated into a single-cell suspension by trituration through syringes. The cells were plated onto a Poly-D-lysine 75T bottle and grown in Dulbecco’s modified Eagle’s media (DMEM) supplemented with 10% inactivated fetal bovine serum (FBS) and 1% antibiotics (P/S, penicillin/streptomycin; NCM). The next day, cells were washed with DMEM to remove debris, and the media were changed twice per week. After 10 days, loosely attached microglia were removed from underlying astrocytes by shaking the bottle at 180 RPM for 30 min at 37 °C. Cells were collected, replated onto dishes in the original culture medium, and allowed to adhere overnight. The next day, the media were replaced with new media. Primary microglia were treated with 5uM Aβ_1-42_ (GL Biochem Ltd., Shanghai, China) for 18 h to construct a cell model of Alzheimer’s disease.

#### 4.9.2. IF Staining

Brain tissues were fixed with 4% paraformaldehyde (PFA) for 10 min. For IF staining, the tissue slices were permeabilized with 0.1% Triton X-100 for 10 min, blocked with 1% BSA for 1 h at 37 °C, and then incubated with antibodies against AQP4 (Servicebio, Wuhan, China, GB12529, 1:200), CD31 (R&D, AF3628, 1:200), and Aβ (Abcam, ab201060, 1:200) at 4 °C overnight. The next day, tissue slices were incubated with fluorescent secondary antibodies for 1 h at room temperature. The nuclei were stained with DAPI Fluoromount-G™ (Yeasen, Shanghai, China, 36308ES20) for 10 min prior to imaging.

After the same pre-treatment as before, the cell slides were incubated with antibodies against CD163 (ProteinTech Group, Chicago, IL, USA, Cat No: 16646-1-AP, 1:400) and iba-1 (WAKO, 011-27991) and the next steps are also consistent with the above. Representative regions and cells were selected and photographed.

#### 4.9.3. RNA Extraction and RT-qPCR

RNA was extracted from the cerebral cortex tissues using NGzol reagent (Invitrogen, Waltham, MA, USA, 15596018) and reverse-transcribed to cDNA using the Hifair^®^ II 1st Strand cDNA Synthesis Kit (Yeasen Biotechnology, Shanghai, China, 11119ES60). Primer sequences were obtained from PrimerBank (https://pga.mgh.harvard.edu/primerbank; accessed on 20 October 2023) and synthesized at Synbio Technologies (Suzhou, China) (Table 3). RT-qPCR was performed using the Hieff^®^ qPCR SYBR Green Master Mix (High Rox Plus; Yeasen Biotechnology, 11203ES03) on a 7900HT Fast real-time PCR System. PCR amplification was conducted in triplicate for each sample, and the expression of target genes was normalized to GAPDH. Relative expression was determined using the 2^−ΔΔCt^ method.

### 4.10. Statistical Analysis

Differences between two groups were assessed using the Student’s *t*-test. For comparisons involving three groups, in the analysis of normally distributed continuous variables, group characteristics were compared using a one-way analysis of variance (ANOVA), followed by a Bonferroni multiple-comparison post-hoc test. In cases where the assumption of equal variance was violated, Tamhane’s T2 test was applied [59]. The Kruskal-Wallis test for continuous variables with skewed distribution and the chi-square test for categorical variables were also used. All clinical statistical analyses were performed using IBM SPSS Statistics (version 26.0), and experimental data analyses were performed using GraphPad Prism 9 (GraphPad Software Inc., La Jolla, CA, USA). A two-tailed *p*-value of <0.05 was considered statistically significant.

## 5. Conclusions

Utilizing the ADNI database and combining the comprehensive reports showed that CD163 is related to small-vessel injury in AD patients. Subsequent validation studies revealed a potential correlation between CD163 and small-vessel injury in the cortical tissue of 5×FAD mice. Additionally, we conducted an analysis of the correlation among the expression levels of feature-selection genes, the abundance of infiltrating immune cells, and DTI indices in brain regions affected by small-vessel injury. These insights may contribute to an enhanced comprehension of the pathological mechanisms of MCI, specifically in relation to small-vessel injury. Furthermore, we suggest that high-risk individuals take measures aimed at preventing microvascular damage, including the control of vascular risk factors such as hypertension, diabetes, hyperlipidemia, etc., as well as the promotion of healthy lifestyles, especially limiting alcohol consumption and embracing an anti-inflammatory dietary regimen.

In conclusion, this research may serve as a predictive tool for estimating the conversion probability from being CN to having MCI and provide novel insights into the physiological mechanisms of early AD with small-vessel injury.

## Figures and Tables

**Figure 1 ijms-25-02293-f001:**
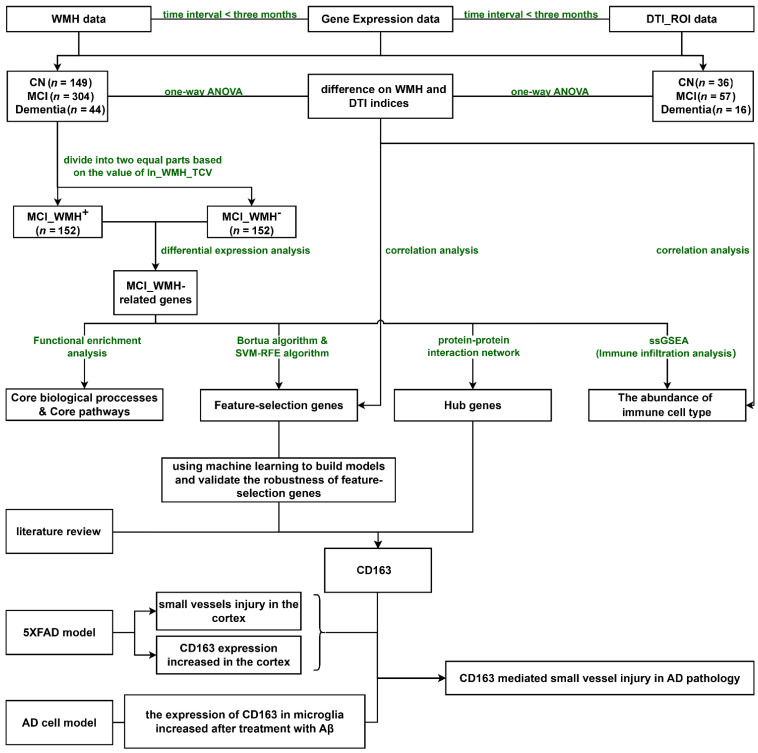
The study workflow chart. Abbreviations: WMH, white-matter hyperintensities; DTI_ROI, diffusion tensor imaging region of interest; CN, cognitively normal; MCI, mild cognitive impairment; AD, Alzheimer’s disease; ln_WMH_TCV, natural logarithm of standardized white-matter hyperintensities (WMH) to total cerebrum cranial volume (TCV); MCI_WMH+, mild cognitive impairment with severe white-matter hyperintensities, whose values of ‘ ln_WMH_TCV’ were above the median; MCI_WMH–, mild cognitive impairment with no or mild white-matter hyperintensities, whose values of ‘ ln_WMH_TCV’ were below the median.

**Figure 2 ijms-25-02293-f002:**
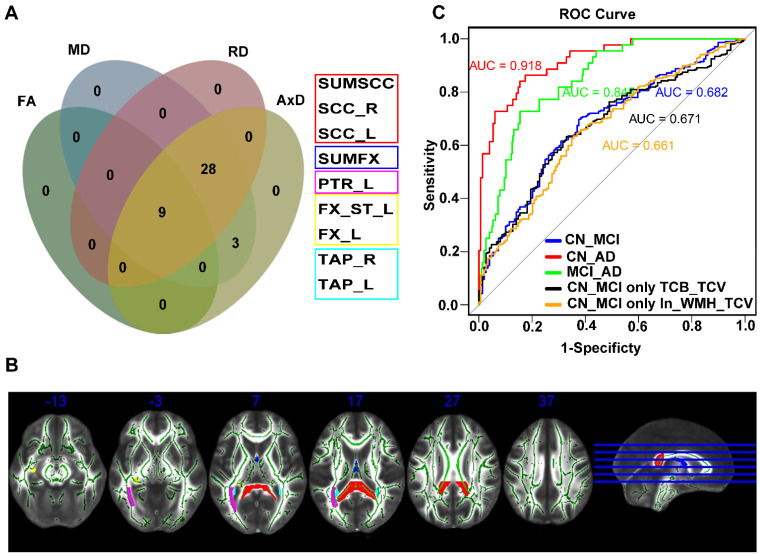
The difference among CN, MCI, and dementia in DTI indices and the prediction accuracy of ln_WMH_TCV and TCB_TCV in predicting AD progression: (**A**) The intersection of difference in FA, MD, RD, and AxD indices were considered the most fragile fiber bundles; (**B**) The predictive effectiveness of cerebral atrophy and small-vessel injury at different stages of AD diagnosis; (**C**) different cross-sectional views of the fragile fiber bundles. Red: bilateral splenium of the corpus callosum (SUMSCC, SCC_R, SCC_L), blue: bilateral fornix (SUMFX), violet: posterior thalamic radiation_left (PTR_L), yellow: fornix (cres)/stria terminalis_left (FX_ST_L, FX_L), cyan: tapetum (TAP_R, TAP_L). Abbreviations: FA, fractional anisotropy; MD, mean diffusivity; RD, radial diffusivity; AxD, axial diffusivity, TAP_R, tapetum right; TAP_L, tapetum left; SUMSCC, bilateral splenium of the corpus callosum; SUMFX, bilateral fornix; SCC_R, splenium of corpus callosum right; SCC_L, splenium of corpus callosum left; PTR_L, posterior thalamic radiation left; FX_ST_L, fornix (cres)/stria terminalis left; FX_L, fornix left; CN, cognitively normal; MCI, mild cognitive impairment; AD, Alzheimer’s disease; ROC, receiver operating characteristic; AUC, area under the curve; TCB_TCV, standardized total cerebrum brain volume (TCB) to total cerebrum cranial volume (TCV); ln_WMH_TCV, natural logarithm of standardized white-matter hyperintensities (WMH) to total cerebrum cranial volume (TCV).

**Figure 3 ijms-25-02293-f003:**
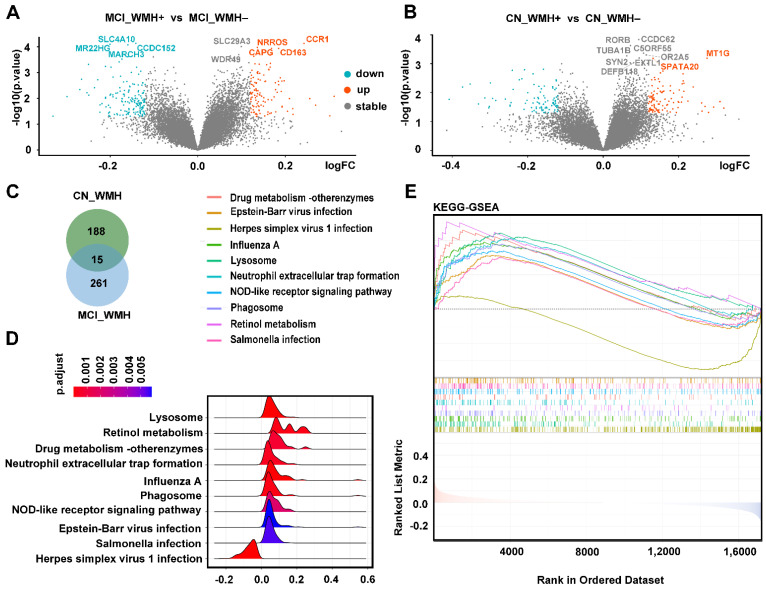
Identification of DEGs and functional enrichment analysis: (**A**) Volcano plot of DEGs constructed using the fold-change values (0.12) and *p*-value (0.05); red-orange color dots represent genes upregulated in MCI_WMH+, gray dots represent genes not differing significantly between MCI_WMH+ and MCI_WMH–, and cyan dots represent genes downregulated in MCI_WMH+. (**B**) Volcano plot of DEGs constructed using the fold-change values (0.12) and *p*-value (0.05); red-orange color dots represent genes upregulated in CN_WMH+ group, gray dots represent genes not differing significantly between CN_WMH+ and CN_WMH– groups, and cyan dots represent genes downregulated in CN_WMH+ group. (**C**) Venn plot. Only fifteen genes shared between WMH-related genes in CN patients and those with MCI. (**D**,**E**) The results of GSEA analysis of KEGG in MCI_WMH+ samples. Abbreviations: MCI_WMH+, mild cognitive impairment with severe white-matter hyperintensities; MCI_WMH–, mild cognitive impairment with no or mild white-matter hyperintensities; CN_WMH+, cognitively normal with severe white-matter hyperintensities; CN_WMH–, cognitively normal with no or mild white-matter hyperintensities; KEGG_GSEA, gene set enrichment analysis (GSEA) of Kyoto Encyclopedia of Genes and Genomes (KEGG) pathways; DEGs, differentially expressed genes.

**Figure 4 ijms-25-02293-f004:**
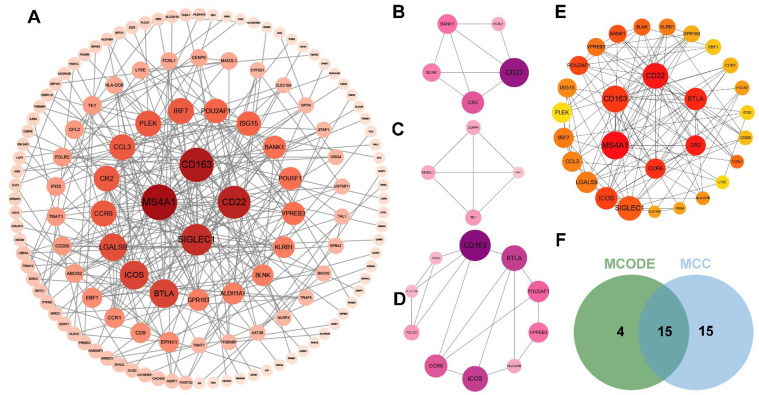
Screening of closely related genes and hub genes of WMH-related genes in MCI group using cytoHubba and MCODE plugins: (**A**) Macroscopic display of PPI networks for all DEGs of WMH-related genes in MCI group, with a redder color indicating a higher degree score; the gene nodes in the topological characteristics of this PPI network were ranked in descending order of degree value (**B**–**D**), with a deeper purple color indicating a higher score. The top three modules’ genes are filtered by MCODE; (**E**) A redder color indicates a higher score and a yellower color indicates a lower score. The hub genes are filtered by the MCC for the top 30 genes; (**F**) Venn plot the common genes are both filtered by MCODE and MCC. Abbreviations: MCODE, molecular complex detection; MCC, maximal clique centrality.

**Figure 5 ijms-25-02293-f005:**
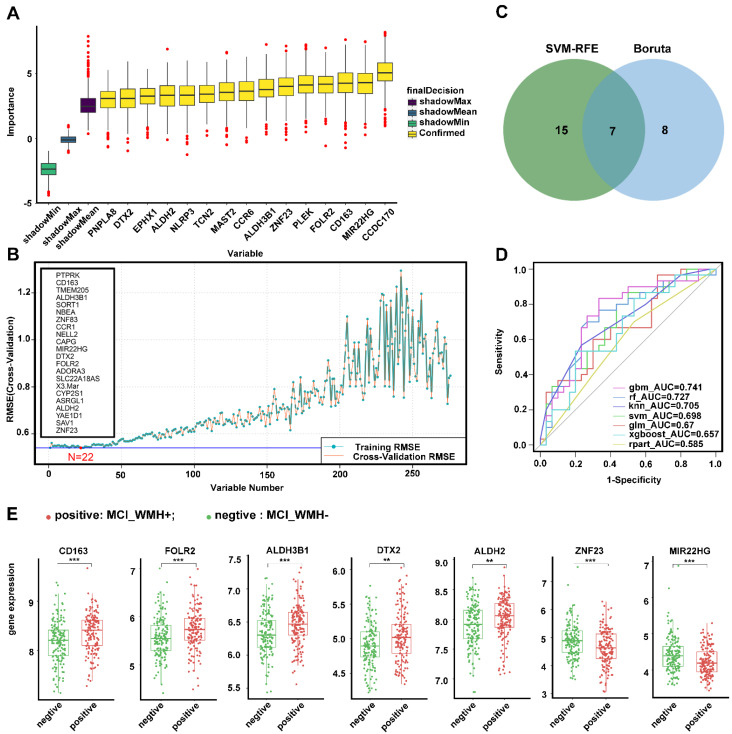
Screening feature-selection genes from DEGs between mild and severe small-vessel injury in MCI group: (**A**) 15 feature-selection genes were screened by Boruta algorithm; (**B**) 22 feature-selected genes were screened by SVM-RFE algorithm; (**C**) Venn plot, seven variables including CD163, FOLR2, ALDH3B1, DTX2, ALDH2, ZNF23, and MIR22HG intersected by Boruta and SVM-RFE algorithms; (**D**) the ROC curve of seven machine learning models, and the AUC value represents the model predictive effectiveness in testing set; (**E**) the expression level of feature-selection genes CD163, FOLR2, ALDH3B1, DTX2, ALDH2, ZNF23, and MIR22HG in the MCI between mild and severe small-vessel injury. CD163, *p*-value = 1.2 × 10^−4^; FOLR2, *p*-value = 6.4 × 10^−4^; ALDH3B1, *p*-value = 4.7 × 10^−4^; DTX2, *p*-value = 1.8 × 10^−3^; ALDH2, *p*-value = 3 × 10^−3^; ZNF23, *p*-value = 5.1 × 10^−4^; MIR22HG, *p*-value = 8.6 × 10^−5^. ** *p* < 0.01; *** *p* < 0.001. Abbreviations: SVM-RFE, support vector machine recursive feature elimination; RMSE, root mean square error; AUC, area under the curve; gbm, gradient boosting machine; rf, random forest; KNN, k-nearest neighbors; SVM, support vector machine; GLM, generalized linear model; XGboost, extreme gradient boosting; rpart, recursive partition tree.

**Figure 6 ijms-25-02293-f006:**
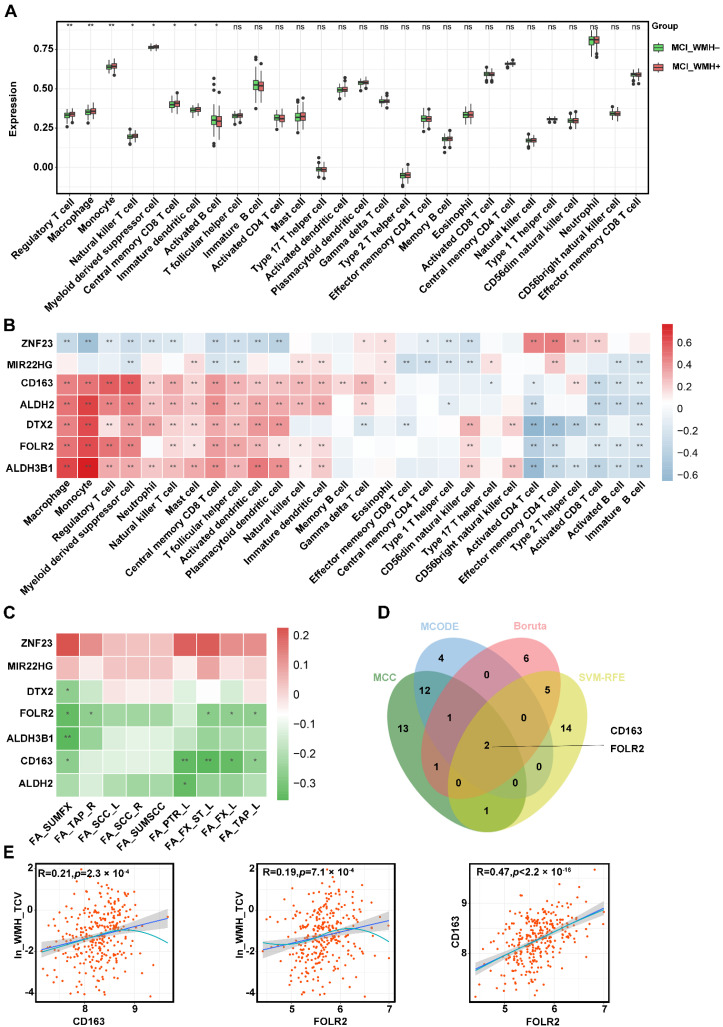
Alterations in the abundance of immune cells in MCI group with severe small-vessel injury (MCI_WMH+) and correlation analysis between hub genes with the abundance of immune cell and FA values: (**A**) Estimated proportions of 28 immune cell types between two groups in MCI group; (**B**) Correlation analysis of hub genes with different immune cell types; (**C**) Correlation analysis of hub genes with differential brain-area FA values; (**D**) Venn plot, two genes (CD163 and FOLR2) intersected by Feature-selection algorithms and PPI. (**E**) The correlation between CD163, FOLR2, and ln_WMH_TCV. * *p* < 0.05; ** *p* < 0.01. Abbreviations: FA, fractional anisotropy; SUMFX, bilateral fornix; TAP_R, tapetum right; SCC_L, splenium of corpus callosum left; SCC_R, splenium of corpus callosum right; SUMSCC, bilateral splenium of the corpus callosum; PTR_L, posterior thalamic radiation left; FX_ST_L, fornix (cres)/stria terminalis left; FX_L, fornix left; TAP_L, tapetum left; ln_WMH_TCV, natural logarithm of standardized white-matter hyperintensities (WMH) to total cerebrum cranial volume (TCV).

**Figure 7 ijms-25-02293-f007:**
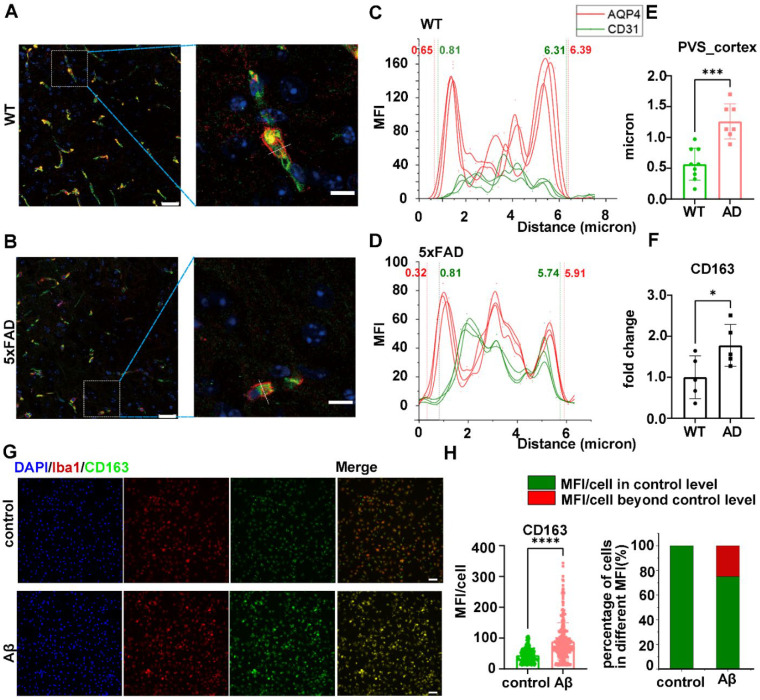
AD pathology leads to small-vessel injury and elevates CD163 expression. Immunofluorescence staining results of (**A**) 5×FAD cerebral cortex (left scale bar = 25 um; right scale bar = 10 um) and (**B**) WT cerebral cortex, green: CD31; red: AQP4; magenta: Aβ; blue: DAPI. (**C**–**E**) Calculation of the width of the perivascular spaces and the comparation between WT and 5×FAD. (**F**) rt-qPCR results from cortical tissues exhibited an upregulation of CD163 expression in 5×FAD mice. (**G**) Immunofluorescence staining: control vs. Aβ-treated rat primary microglia (scale bar = 50 um), green: CD163; red: iba-1; blue: DAPI. (**H**) Immunofluorescence staining results of CD163 showing the MFI and the percentage of cells in different MFIs. * *p* < 0.05; *** *p* < 0.001; **** *p* < 0.0001. Abbreviations: MFI, mean fluorescence intensity; Aβ, amyloid beta.

**Table 1 ijms-25-02293-t001:** Demographic characteristics, neuropsychological tests, and neuroimaging indices.

	CN (*n* = 149)	MCI (*n* = 304)	Dementia (*n* = 44)	*p*
Demographic characteristics				
Male, *n* (%)	68 (45.60%)	163 (53.60%)	25 (56.80%)	0.090
Age, mean (SD)	73.45 (5.99) *^b^*	71.46 (7.42) *^ac^*	75.23 (9.20) *^b^*	0.001
Education, mean (SD)	16.68 (2.53) *^c^*	16.10 (2.64)	15.50 (2.71) *^a^*	0.013
APOEε4 carriers, *n* (%)	105/38/6 (70.50%/25.50%/4.00%) *^bc^*	175/106/23 (57.60%/34.90%/7.60%) *^ac^*	11/26/7 (25.00%/59.10%/15.90%) *^ab^*	<0.001
Neuropsychological tests				
MMSE, mean (SD)	29.03 (1.21) *^bc^*	28.11 (1.62) *^ac^*	22.52 (3.04) *^ab^*	<0.001
FAQ, mean (SD)	0.16 (0.60) *^bc^*	2.44 (3.56) *^ac^*	13.57 (7.20) *^ab^*	<0.001
MoCA, mean (SD)	25.72 (2.16) *^bc^*	23.52 (3.07) *^ac^*	17.19 (4.93) *^ab^*	<0.001
CDRSB, mean (SD)	0.03 (0.14) *^bc^*	1.38 (0.86) *^ac^*	4.64 (1.77) *^ab^*	<0.001
Neuroimaging				
ln_WMH_TCV, mean (SD)	−1.34 (1.16)	×1.23 (1.20)	−0.99 (1.10)	0.209
TCB_TCV, mean (SD)	77.71 (2.52) *^c^*	77.56 (3.02) *^c^*	74.26 (2.39) *^ab^*	<0.001
T_hippo_TCV, mean (SD)	0.56 (0.05) *^bc^*	0.54 (0.07) *^ac^*	0.46 (0.07) *^ab^*	<0.001

The χ2-test for gender and APOEε4, and one-way ANOVA were performed to assess group comparison for other indices, and Bonferroni correction for homoscedasticity and Tamhane’s T2 test for heteroscedasticity were carried out; *p* < 0.05 was considered to be statistically significant. *a:* Significantly different from NC; *b:* Significantly different from MCI; *c:* Significantly different from dementia. Abbreviations: CN, cognitively normal; MCI, mild cognitive impairment; SD, standard deviation; MMSE, mini mental state examination; FAQ, functional activities questionnaire; MoCA, Montreal cognitive assessment; CDRSB, clinical dementia rating scale sum of boxes; ln_WMH_TCV, natural logarithm of standardized white-matter hyperintensities (WMH) to total cerebrum cranial volume (TCV); TCB_TCV, standardized total cerebrum brain volume (TCB) to total cerebrum cranial volume (TCV); T_hippo_TCV, standardized segmented total hippocampi volume (T_hippo) to total cerebrum cranial volume (TCV).

**Table 2 ijms-25-02293-t002:** Logistic regression analysis of diagnosis during AD continuum.

	Odds Ratio (95% CI)	*p*	C-Statistics
CN vs. MCI ^[1]^			
ln_WMH_TCV	1.262 (1.041–1.537)	0.019	0.6614
CN vs. MCI ^[2]^			
TCB_TCV	0.887 (0.809–0.970)	0.010	0.671
CN vs. MCI ^[3]^			
ln_WMH_TCV	1.278 (1.053–1.559)	0.014	0.682
TCB_TCV	0.882 (0.804–0.966)	0.007	
CN vs. Dementia ^[4]^			
ln_WMH_TCV	1.322 (0.874–2.083)	0.202	0.918
TCB_TCV	0.505 (0.385–0.634)	<0.001	
MCI vs. Dementia ^[5]^			
ln_WMH_TCV	1.081 (0.759–1.548)	0.668	0.845
TCB_TCV	0.657 (0.554–0.768)	<0.001	

Model [1] and [2] only included ln_WMH_TCV or TCB_TCV for univariate logistic regression analysis, Model [3], [4], and [5] included two dependent variables, ln_WMH_TCV and TCB_TCV, for multivariate logistic regression. All logistic regression models were adjusted for age, gender, APOEε4, and education. Abbreviations: 95% CI, 95% confidence interval; C-statistics, concordance statistic; CN, cognitively normal; MCI, mild cognitive impairment; ln_WMH_TCV, natural logarithm of standardized white-matter hyperintensities (WMH) to total cerebrum cranial volume (TCV); TCB_TCV, standardized total cerebrum brain volume (TCB) to total cerebrum Cranial Volume (TCV).

**Table 3 ijms-25-02293-t003:** The RT-PCR primer.

Gene	Primer
CD163 (mouse)	CD163-F AATCACATCATGGCACAGGTCACC CD163-R TCGTCGCTTCAGAGTCCACAGG
GADPH (mouse)	GAPDH-F GGCAAATTCAACGGCACAGTCAAG GAPDH-R TCGCTCCTGGAAGATGGTGATGG

## Data Availability

All data is available from ADNI (https://adni.loni.usc.edu/).

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
