# Peer review of "CD163-Mediated Small-Vessel Injury in Alzheimer’s Disease: An Exploration from Neuroimaging to Transcriptomics"

_ijms, 2024, doi:10.3390/ijms25042293_

Round 1
Reviewer 1 Report
Comments and Suggestions for Authors
The manuscript entitled “CD163 mediated small vessel injury in Alzheimer’s disease: An 2 exploration from neuroimaging to transcriptomic " by Chen et al., is potentially an important study, performed and connected finely. The authors have used various parameters to test their hypothesis. The following are some points that the authors should consider.
- The workflow chart in Figure 1 should be explained by the authors
- The authors double-check the statistical analysis in Figure 5(E), " The expression level of feature-selection genes".
- When comparing CD163 expression in Aβ treatment to control, did the authors check the percentage increased?
- In figure 7(G) Immunofluorescence staining, authors should double-check the merged image of Aβ-treated rat.
Thanks
Reviewer 2 Report
Comments and Suggestions for Authors
Tha manuscript addresses a topic of interest, namely small vessel injury in MCI and early stage AD. Considering the improving knowledge on pathological early signatures of this disease would enhanced comprehension of the pathological mechanisms of MCI and positively impact patient stratification. Currently no effective drugs are available to treat AD, hence, outcomes of this ma work may also help identifying new molecular targets for drug discovery, in my opinion the manuscript own sufficient interest for publication on the IJMC.
The work is clearly presented as well as the main outcomes. English language is good. Minor mystypos can be easily revised.
Limitations of the work are critically presented and discussed.
Comments on the Quality of English LanguageThe manuscript is clearly written. English style is fluent and just really few mistypos have been detected.
